# Understanding the Interaction of Lignosulfonates for the Separation of Molybdenite and Chalcopyrite in Seawater Flotation Processes

**DOI:** 10.3390/polym14142834

**Published:** 2022-07-12

**Authors:** Consuelo Quiroz, Romina Murga, Juan David Giraldo, Leopoldo Gutierrez, Lina Uribe

**Affiliations:** 1Escuela de Ingeniería Civil de Minas, Universidad de Talca, Curicó 334000, Chile; cquiroz14@alumnos.utalca.cl (C.Q.); romina.murga@utalca.cl (R.M.); 2Instituto de Acuicultura, Universidad Austral de Chile, Puerto Montt 5480000, Chile; juan.giraldo@uach.cl; 3Departamento de Ingeniería Metalúrgica, Universidad de Concepción, Concepción 4030000, Chile; lgutierrezb@udec.cl; 4Centro de Recursos Hídrico para la Agricultura y la Minería (CRHIAM), Universidad de Concepción, Concepción 4030000, Chile

**Keywords:** lignosulfonates, molybdenite flotation, chalcopyrite, seawater

## Abstract

The selective separation of molybdenite from copper sulfide concentrate in flotation process is realized using sodium hydrosulfide (NaHS) to depress the chalcopyrite and permit only the flotation of the molybdenite. However, this reagent is a highly toxic and flammable gas. The objective of this research was to study the feasible application of commercial lignosulfonates (LSs) in the separation by froth flotation process of molybdenite and chalcopyrite in seawater to present a novel application for LSs, as well as an alternative reagent to sodium hydrosulfide (NaHS). To achieve this, microflotation, absorbance tests and zeta potential measures were performed at pH 8 in seawater and 0.01 M NaCl. The results obtained in this study showed that it is possible to achieve selective separation of copper and molybdenum in both aqueous media due to high depressant effect of molybdenite and low depression of chalcopyrite in microflotation tests at 10 ppm of LSs, when the collector, PAX, is added prior to the addition of LSs. Absorbance study and zeta potential measurements showed that LSs adhere more to the molybdenite surface in seawater than in freshwater. This is related to the high ionic charge of the media that influences a greater interaction of LSs with the mineral surface.

## 1. Introduction

Mining activity is one of the most important economic areas in Chile, due to the large number of mineral reserves it possesses. Within its metal mining, the production of copper and molybdenum stands out, the latter being obtained as a byproduct of copper production. Commonly, the selective separation of molybdenite from copper sulfide concentrates (mainly composed of chalcopyrite and chalcocite), is the second stage of a froth flotation process, where the copper and the molybdenum were collectively separated from the gangue in a first stage. In this selective second stage, the sodium hydrosulfide (NaHS) is frequently used to remove the collector adsorbed on the chalcopyrite surface in the first stage, depressing the chalcopyrite and allowing only the flotation of the molybdenite [1].

However, the use of NaHS as a depressant of chalcopyrite is highly hazardous and contaminating because it easily transforms into hydrogen sulfide ($H_{2}$S), a toxic and flammable gas [2] which is harmful to both humans and the environment [3]. In addition, there is a need to replace such toxic and hazardous depressants with more environmentally friendly chemicals, considering that the control regulations against environmental pollution are becoming stricter [3]. Considering this, to avoid its use, several selective depressants of molybdenite have been proposed in order to allow the flotation of the chalcopyrite and the inverse recovery of the molybdenite [4]. One of these depressant agents is the lignin, specifically its derivatives, the Lignosulfonates (LS) [5,6]. These are renewable resources with low applications in the industry and most of them are disposed as waste, causing a negative environmental impact. For that reason, taking full advantage of lignosulfonates is important for both economic and environmental reasons [7]. Previous work using 0.01 M NaCl as aqueous solution showed that commercial lignosulfonates can selectively depress hydrophobic molybdenite without affecting chalcopyrite recovery, if a long-chain collector such as potassium amyl xanthate is used to prevent the interaction of mineral with the LS. Then, a molybdenite depression higher than 60% and a recovery of chalcopyrite over 95% were obtained when the flotation process was performed at pH 9 in the presence of a high concentration of LSs [8].

The potential selective separation of both minerals using these lignosulfonates as depressants of molybdenite in freshwater us gave the motivation to study this selective froth flotation process replacing the freshwater with seawater (SW). In Chile, some copper flotation plants use seawater, and have done so for decades [9]. Examples include the Cu-Mo processing plants “Las Luces” in Taltal (Antofagasta) [10], “Carolina Michilla S. A.” in Mejillones, and “Esperanza” in Sierra Gorda [9]. At present, there is a great shortage of water resources in mining areas, so it is necessary to look for other water sources that allow the processing of the ore, and seawater is the most common alternative. Specifically, in Chile, for the year 2029 a consumption of 14.53 $m^{3}$/s of freshwater and 10.83 $m^{3}$/s of seawater is projected in mining activity. Compared to 12.98 $m^{3}$/s and 3.28 $m^{3}$/s consumed in 2018, this represents an increase of 12% and 230%, respectively [11]. The great challenge in the direct use of seawater is in studying the best operating conditions and proposing new reagents that help to obtain primary metal (copper) concentrates and byproducts, such as molybdenum with characteristics equal to or superior to those obtained in flotation with freshwater. This is because seawater contains large amounts of dissolved ions that mainly affect the natural floatability of molybdenite at basic pH, due to the presence of divalent cations that generate products of hydrolysis and precipitation of calcium and magnesium salts [12].

The objective of this research was to study the feasible application of commercial LSs in the separation by froth flotation process of molybdenite and chalcopyrite in seawater to present a novel application for LSs, as well as an alternative reagent to sodium hydrosulfide (NaHS). In order to achieve this, the chalcopyrite and molybdenite recovery were evaluated through microflotation tests in seawater and 0.01 M NaCl, and the absorbance tests were carried out to analyze the degree of lignosulfonate adsorption on the minerals surface. Finally, the electrophoretic mobility measurements were realized to study the interaction of molybdenite with the LSs.

## 2. Materials and Methods

### 2.1. Reagents

Potassium amyl xanthate (PAX) supplied by ORICA (Santiago, Chile) was used as the collector and previously purified according to the methodology described by Montalti [13], using acetone (99.8%, MERCK, Chile) and ethyl ether (99.7%, MERCK). Methyl isobutyl carbinol (MIBC) purchased from MERCK was used as frother. Hydrochloric acid (37%), sodium hydroxide pellets (≥97%, and sodium chloride (99.99%) purchased from MERCK were used to prepare the pH modifiers and the 0.01 M NaCl solution, respectively. Seawater was obtained from the coastal area of the Maule Region and was filtered to remove any organic traces. Type II water (1 MΩ·cm) was used to prepare the different solutions.

The sodium and calcium lignosulphonates (Na-LS or Ca-LS) were supplied by Sigma-Aldrich (Santiago, Chile) and characterized by elemental analysis, molecular masses, and Fourier Transform Infrared Spectroscopy (FTIR) using a Nexus WQF-510 A FTIR spectrophotometer.

### 2.2. Chalcopyrite

The ore was supplied by Ward’s Natural Science. The sample was ground using a ceramic mortar to obtain 250 g of chalcopyrite with a final particle size −105/+38 μm, which was characterized using a multi-elemental analysis Wavelength Dispersive X-ray Fluorescence (WDXRF) S8-TIGER (Bruker, Billerica, MA, USA) spectrometer with a Rh excitation source and a X-ray Difractometer (XRD) MiniFlex 600 (Rigaku, Japan).

Table 1 shows the elemental composition of the ore obtained by X-ray Fluorescense (XRF) Niton XL3t (Thermo Fisher Scientific, Waltham, MA, USA), which indicates that the chalcopyrite sample was composed of 33.06%w copper, 29.33%w iron, and 29.20%w sulfur. Other elements, such as silicon, aluminum, magnesium, sodium, calcium, and potassium were below 1.00%w. With these results, a purity of 95.46% chalcopyrite was obtained. Furthermore, X ray diffraction (XRD) analysis showed that the ore was mainly composed of chalcopyrite and quartz, which is consistent with what was observed by XRF analysis.

### 2.3. Molybdenite

The molybdenite sample was obtained in the form of a concentrate. The sample had to be manually ground in a ceramic mortar and then sieved to obtain a particle size −105/+38 μm. Subsequently, the sample was purified following the procedure established by Gutierrez et al. [8]

Table 2 presents the elemental composition of the sample analyzed by XRF, which shows that the concentrate obtained is composed mainly of 45.76%w molybdenum, 39.87%w sulfur, and 2.40%w silicon and other elements such as Cu, Fe, Al, and Cl with a concentration less than 1.00%. With these results, a purity of 76.34% molybdenite was obtained. On the other hand, XRD analysis showed that the concentrate is mainly composed of molybdenite pyrite and quartz.

### 2.4. Micro-Flotation Procedure

A 140 mL Partridge–Smith glass cell was used to perform the micro-flotation tests. The assays were performed to evaluate the depression of molybdenite with the reagents under study, i.e., sodium and calcium lignosulfonates, and the effect of the presence of these on the recovery of chalcopyrite. Additionally, the order of addition of reagents in the chalcopyrite flotation was evaluated. The tests were carried out in seawater and freshwater at pH 8. It is important to mention that each micro-flotation test was performed in duplicate for the validation of results.

The procedure started by suspending 1 g of mineral (molybdenite or chalcopyrite) in 110 mL of solution (0.01 M NaCl or seawater) for 2 min at the required pH. Then, lignosulfonates were added to the suspension and conditioned for additional 10 min. Finally, MIBC + PAX was added in the case of chalcopyrite or only MIBC in the case of molybdenite, at a fixed concentration (25 ppm PAX, 15 ppm MIBC), and conditioned for 5 more minutes. Once this was achieved, the flotation process was carried out, and the pulp was transferred to the Partridge–Smith cell, where it was conditioned for 20 s, and then the airflow in the cell began (80 mL/min). Flotation was performed for 2 min with manual scrapping every 10 s. In the end, the concentrate and the tail obtained in each test were filtered and dried. The percentage of recovery was determined according to Equation (Equation 1).
(1)$$\mathrm{Recovery}\left(\%\right)=\frac{\mathrm{Mass}\mathrm{of}\mathrm{concentrate}(g)}{\mathrm{Mass}\mathrm{of}\mathrm{concentrate}(g)+\mathrm{Mass}\mathrm{of}\mathrm{tail}(g)}*100$$

To evaluate the order of addition of reagents in the recovery of chalcopyrite, the lignosulfonates and PAX were added at the same time or separately, according to Table 3.

### 2.5. UV-Visible Spectrophotometer Test

The UV-Visible spectrophotometer used was the SPECORD® 50 PLUS (Analytic Jena) equipped with the ASpect UV basic software. The absorbance tests were carried out to analyze the degree of lignosulfonate adsorption on the surface of molybdenite and chalcopyrite minerals in the different aqueous media studied. To measure the amount of lignosulfonate adsorbed on the surface of the mineral, the amount of lignosulfonate remaining in solution (measure by absorbance tests) was subtracted from the amount of lignosulfonate initially added.

Firstly, the absorption spectra of each pure reagent were analyzed separately (Na-LS, Ca-LS, PAX, and MIBC), in freshwater and seawater, to analyze the characteristic bands of each reagent. Then, the absorption spectra of the mixtures of reagents in the different media were analyzed to determine possible interaction between them, and later, to analyze the adsorption of the lignosulfonates on the minerals. In these tests, the previously described conditioning process was followed, and aliquots (3 mL) of the filtered solutions were analyzed.

To determine the concentration of lignosulfonates, present in solution, the following calibration curves expressed as linear Equations (Equation 2)–(Equation 5) were obtained using freshwater and seawater as solvents (see Table 4). The calibrations curves were elaborated using five (Equation 5) standards of each lignosulfonate (Na-LS or Ca-LS) in the concentration range of 1–50 ppm (triplicate measures).

### 2.6. Electrophoretic Mobility Measurements

Zeta potential measurements using the Litesizer 500^®^ (Anton Paar, Austria) were realized to study the interactions between molybdenite and lignosulfonates. A suspension of molybdenite (0.05%) was prepared by mixing 0.1 g of solid in 200 mL of solution freshwater (0.01 M NaCl) or seawater. The samples were obtained via the following procedure. First, the pH of the solution was adjusted at pH 8 using sodium hydroxide (NaOH) and hydrochloric acid (HCl). After this, the mineral was added into the solution and mixed for 2 min. Subsequently, an aliquot of lignosulfonate was added to the pulp according to the desired concentration (between 0 and 50 ppm) and the solution was conditioned for 10 min. Once this process was complete, 0.4 mL of solution was used for the zeta potential tests. Each condition was studied in triplicate in order to determine the standard deviation of these measures. It is important to note that the conductivity of NaCl 0.01 M and seawater were registered when the pH was adjusted, and these values corresponded to $1.1\times 10^{3}\;\mu$S/cm and 52 mS/cm, respectively.

## 3. Results

### 3.1. LSs Characterization

Table 5 shows that the molecular weights (Mw) and molecular number (Mn) of Na-LS and Ca-LS were 54,000 and 18,000 g/mol and 7268 and 2500 g/mol, respectively. The sodium concentrations were 20.58% (Na-LS) and 5.16% (Ca-LS) and calcium concentrations were 1.44% (Na-LS) and 16.53% (Ca-LS). Figure 1 shows the FTIR spectra of Na-LS and Ca-LS. The peak displayed at around 1600–1650 cm${}^{-1}$, corresponding to C=O stretches, reveals the presence of carboxylic groups. The band displayed at 3100–3700 cm${}^{-1}$ corresponding to O-H stretches most probably relates to phenolic groups and water. The peaks displayed in the range 1100–1500 cm${}^{-1}$ corresponding to S=O stretches, are associated with the sulfonates and sulfonic acids, among others.

### 3.2. Flotation of Molybdenite and Chalcopyrite in the Presence of Lignosulfonates

Figure 2 shows the recovery of molybdenite and chalcopyrite, as a function of the concentration of Ca-LS in freshwater and seawater, respectively, using 15 ppm MIBC for both minerals and 25 ppm PAX as chalcopyrite collector at pH 8. In the absence of Ca-LS (0 ppm), it is possible to observe that the saline environment affects the flotation of both minerals.

On the other hand, in the presence of Ca-LS, it can be observed that the recovery of both minerals decreases as their concentration increases. A significant depression of molybdenite was noted at low concentrations of Ca-LS. At 10 ppm Ca-LS, a recovery of 7% molybdenite and 80% chalcopyrite was obtained in freshwater and a recovery of molybdenite 5% and chalcopyrite 61% in seawater. The above shows a similar trend in the recovery of molybdenite in both media, while in the case of chalcopyrite recovery, it was observed that it was affected to a greater extent in the seawater and in the presence of Ca-LS.

Figure 3 shows the recovery of molybdenite and chalcopyrite as a function of Na-LS concentration in freshwater and seawater. A similar trend to that obtained in the presence of Ca-LS is observed, i.e., there is a high decrease in molybdenite recovery at the lower concentration of lignosulfonate used. However, in the presence of this lignosulfonate, a greater depressing effect on chalcopyrite in freshwater is evident, resulting in a 67% recovery of chalcopyrite at 10 ppm of Na-LS. In seawater, the trend is similar, with a 65% recovery of chalcopyrite. In the case of molybdenite, the difference is low in both media, with 7% and 4% percentage points in freshwater and seawater, respectively.

Figure 4 and Figure 5 show the results of chalcopyrite recovery obtained in absence of LSs and with the addition of LSs in different stages: Adding the LSs before the collector (LSs, PAX+MIBC), after the collector (PAX, LSs+MIBC), and at the same time as the collector (LSs+PAX, MIBC), in the different media studied.

In Figure 4, it is possible to note that the recovery of chalcopyrite in freshwater is significantly higher than in seawater in all three types of addition order. However, the recovery of chalcopyrite was similar to that obtained in absence of lignosulfonate, when PAX was added prior to the addition of Ca-LS, in both media. On the other hand, Figure 5 shows the effect of the order of addition of the reagents in the presence of Na-LS. In this figure, it is possible to observe that the utilization of this lignosulfonate presented lower recoveries of chalcopyrite compared to those obtained with Ca-LS in both media. In addition, the best chalcopyrite recoveries were also obtained when PAX was added first and followed by Na-LS.

### 3.3. UV-Visible Absorbance Tests

#### 3.3.1. UV-Visible Spectra of Reagents in the Absence of Mineral

Figure 6, Figure 7, Figure 8 and Figure 9 present the absorbance spectra obtained for each reagent and its mixtures involved in the microflotation tests (Ca-LS, Na-LS, PAX, MIBC, Ca-LS/MIBC, Ca-LS/PAX, Ca-LS/MIBC/PAX, Na-LS/MIBC, Na-LS/PAX, Na-LS/MIBC/PAX) in both 0.01 M NaCl (Figure 6 and Figure 8) and seawater (Figure 7 and Figure 9).

Regarding the characteristic bands of each reagent in the different media, LSs showed bands with maximum absorbances around 202 nm, 235 nm (shoulder), and 278 nm. In the case of PAX, bands with maximum absorbances at 230 nm and 301 nm were presented, while MIBC did not present any characteristic band in the studied wavelength range. On the other hand, comparing the spectra obtained for each reagent separately, as well as its mixtures, it is noted that there are no displacements in the bands of the mixtures, concerning the bands of each reagent alone in the different mediums studied.

However, in relation to the characteristic bands of the Na-LS and Ca-LS when mixed with PAX, it was observed that there was an increase in the absorbance of the characteristic bands of the Na-LS and Ca-LS when mixed with PAX.

#### 3.3.2. UV-Visible Spectra of Lignosulfonates in the Presence of Mineral

Figure 10 and Figure 11 show the absorbance spectra of Ca-LS and Na-LS, obtained before and after contact with molybdenite and chalcopyrite in 0.01 M (a) and seawater (b) without the use of PAX and MIBC.

The figures show that, in both media, the LSs in contact with chalcopyrite showed higher absorbance than that in contact with molybdenite (see the band at 278 nm). This increase in the absorbance was greater than that obtained in the solution of LSs without minerals.

#### 3.3.3. Zeta Potential of Molybdenite vs. LSs Absorption

Table 6 shows the zeta potential of the molybdenite surface in the absence and presence of 10 ppm LSs and the LSs adsorption calculated at 278 nm, in NaCl 0.01 and seawater.

In the table, it is possible to observe that, in absence of lignosulfonates, the zeta potential of molybdenite surface is less negative in seawater than 0.01 M NaCl. In addition, comparing the results of LSs adsorption obtained in molybdenite in the different aqueous media used, it is possible to note that the LSs are absorbed to a greater extent in seawater than in 0.01 M NaCl.

Finally, comparing the results of zeta potential and adsorption of the different LSs on the molybdenite surface, it is evident that there is a greater adsorption of Na-LS than Ca-LS, in NaCl and in seawater.

## 4. Discussion

Molecular weights (Mw) and molecular numbers (Mn) of commercial LSs have ranges between 18,000 and 54,000 and 2500 and 7268, respectively, which indicate that these lignosulfonates were obtained from hard and soft woods [14]. The LSs studied correspond to anionic polyelectrolytes due to the presence of carboxylic and phenolic groups shown in FTIR spectra. In addition, comparing elemental analysis, it is possible to observe the Ca-LS has higher anionicity than Na-LS, as the cation concentration of Ca-LS was lower than Na-LS.

Microflotation tests showed that, in the absence of lignosulfonates, the saline environment affects the flotation of both minerals due to the high ionic charge of salts present in seawater, with molybdenite being more affected than chalcopyrite. It is known that molybdenite depression at pH 8 can occur due to the adsorption of hydrolysis products of Ca(OH)${}^{+}$, which generate a more hydrated molybdenite surface and therefore contribute to a loss of hydrophobicity [4,8,9,12]. On the other hand, in the presence of the studied LSs (Ca-LS and Na-LS), It can be observed that the presence of these generated a high depressant effect in molybdenite at low concentration (10 ppm) of LSs in 0.01 M NaCl and seawater compared with the depression of chalcopyrite, which is in accordance with our previous work [6,8].However, it was observed that the chalcopyrite recovery was affected by the LSs, when added prior to the PAX collector. This behavior indicates that the previous addition of lignosulfonate before the PAX collector generates an interaction of lignosulfonates with the surface of chalcopyrite, which prevents the subsequent adsorption of the PAX collector and decreases its recovery. In addition, when comparing the recovery of chalcopyrite obtained in the presence of 10 ppm of Ca-LS in 0.01 M NaCl and seawater, it was observed that the flotation of chalcopyrite was less affected in 0.01 M NaCl than in seawater, obtaining a recovery of 85% and 61%, respectively. Meanwhile, when 10 ppm of Na-LS used, the differences in the recovery of chalcopyrite were less noticeable in both media, reaching a recovery close to 65%. In addition, it is important to note that the depressant effect of chalcopyrite in freshwater reverted when 50 ppm of LSs were used. This phenomenon could be attributed to the froth and dispersants characteristics of the lignosulphonates [15,16], which could have been increased as a function of the LSs concentration and may have influenced in the ore recovery. Finally, to compare the chalcopyrite recovery in absence of LSs with those obtained when the LSs were added in a different order in the conditioning stage, it was observed that the order of addition of the reagents can improve the recovery of chalcopyrite. Specifically, the addition of PAX before the addition of LSs allowed us to increase the recovery of chalcopyrite in seawater and 0.01 M NaCl, resulting in a recovery of 80% and 91%, respectively, in the case of Ca-LS and also of 71% and 81% when Na-LS was used. These results suggest that successful selective separation of both minerals is possible under these conditions, preferably using the Ca-LS as the depressant.

On the other hand, absorbance analysis made in the absence of minerals showed no significant interactions between PAX and MIBC, as there are no displacements in the bands of the mixtures, concerning the bands of each reagent alone in the different studied media. Otherwise, concerning the MIBC frothing agent, it can be observed that it does not interact with the LSs and PAX, since its spectra do not show any change or presence of a new band in the studied wavelength range, and there is no significant difference between the media studied for the LSs absorbance spectra, since their maximum absorbances have a minimum variation, either for Ca-LS and Na-LS. However, in relation to the characteristic bands of the Na-LS and Ca-LS when mixed with PAX, it was possible to note that there was an increase in the absorbance of the characteristic bands of the Na-LS and Ca-LS when mixed with PAX. This means that a quantitative analysis of the concentration of these reagents adhered to the surface of the mineral when they are mixed with PAX cannot be performed, since bands at 278 nm (from LSs) and 301 nm (from PAX) results in the sum of its absorbances. Considering this, it is only possible to carry out qualitative monitoring that provides information on the interaction of the LSs and PAX with the minerals.

Furthermore, absorbance analysis realized in the presence of Ca-LS and Na-LS showed that the absorbance spectra of the LSs in contact with chalcopyrite were higher than for those in contact with molybdenite (see the band at 278 nm), indicating a lower adsorption of these reagents on chalcopyrite than on molybdenite. This behavior is related to the results obtained in the microflotation tests, where the depressing effect of the LSs was smaller with chalcopyrite than molybdenite.

In addition, it is important to note that the absorbance spectra in the presence of chalcopyrite was greater than that obtained in the solution of LSs without minerals and higher in freshwater than seawater, indicating that some of the dissolved compounds from the chalcopyrite form a complex that coordinates with the LSs, that the medium affects the interactions between the LSs and the chalcopyrite, and for that reason, a higher depression of chalcopyrite is obtained in seawater. This behavior can be attributed to the LSs, which are highly anionic. At pH 8, the macromolecules of LSs are highly dissociated, i.e., with these groups, they are negatively charged. These punctual negative charges can electrostatically interact with the hydrolysable copper hydroxyl (and other) species from the surface of the chalcopyrite (dissolved) [5,6,17,18,19], and form a soluble complex that absorbs light at 278 nm with more intensity than the salts of LSs alone. It is known that the divalent cations (e.g., Mg${}^{2+}$ and Ca${}^{2+}$) interact with the dissolved LSs macromolecules to form a water-insoluble complex [20,21], which restricts the interaction the LSs with the mineral. For that reason, the absorbance at 278 nm in seawater is lower than in freshwater. Therefore, the addition of PAX before the LSs may allow a reduction in the formation of an insoluble complex in the chalcopyrite surface which decrees the recovery of this mineral.

Finally, it is known that the counterion that accompanies the LS macromolecules (before being dissolved) and the ionic strength of the medium are critical parameters that affect the molecular conformation and solution behavior of the LS macromolecules [21,22], properties intimately linked with its performance as a depressant [23] According to the results of zeta potential and adsorption of the different LSs, on the molybdenite surface, it was possible to observe that the LSs were absorbed to a greater extent in seawater than in 0.01 M NaCl, due to the cations present in seawater (Mg${}^{2+}$, Ca${}^{2+}$, and their hydroxy complexes, mainly) being adsorbed into the molybdenite surface, generating a less negative surface, which decreased the electrostatic repulsion generated between the molecules, contributing favorably to the adsorption. In addition, comparing the results of zeta potential and adsorption of the different LSs on the molybdenite surface, it is evident that there was a greater adsorption of Na-LS than Ca-LS in NaCl and in seawater. This can be related to the degree of anionicity of Na-LS that is lower than Ca-LS [8] and a low anionicity reduces the repulsive forces which to permit a higher adsorption.

The results suggest that the studied commercial LSs achieved the selective separation of molybdenite and chalcopyrite in the flotation process in seawater via the depressing effect of molybdenite of these reagents, as the Ca-LS was the one with the best performance. This separation is possible, at natural pH of seawater, considering the presence of xanthate to float chalcopyrite and at low concentrations of LSs. This alternative reagent could be helpful in the task of reducing the use of sodium hydrosulfide (NaHS), which is highly toxic and flammable, and could contribute to improving the metallurgical performance of the flotation process with seawater. However, more studies are required that consider mixed and real ores in the flotation process to evaluate if the effectiveness of LSs in separating these minerals is maintained.

## 5. Conclusions

Flotations tests showed that it is possible to separate molybdenite and chalcopyrite in seawater using low concentrations of both commercial LSs (Ca-LS and Na-LS). Specifically, when using 10 ppm of LSs with a prior addition of PAX, the molybdenite recovery was near 5% and the chalcopyrite recovery was about 80% (Ca-LS) and 71% (Na-LS).

Zeta potential measurements and adsorption analysis evidenced that the use of seawater contributed to a reduction in the electrostatic repulsion between the LSs and molybdenite molecules, contributing to a higher adsorption of them in the molybdenite surface.

The absorbance analysis performed with Ca-LS and Na-LS and the different minerals indicated a lower adsorption of LS in chalcopyrite than in molybdenite. In addition, it was found that the absorbance spectra of the LS in contact with chalcopyrite were higher than for those in contact with molybdenite (see the band at 278 nm).

Moreover, the absorbance analysis showed that the absorbance spectra in the presence of chalcopyrite were higher than those obtained in the LS solution without minerals, and even these were higher in freshwater than in seawater. This result indicates that some of the dissolved chalcopyrite compounds formed a coordinated complex with the LS, and that the medium increased the interactions between the LS and chalcopyrite. For this reason, the best way to avoid this effect is to add PAX prior to the addition of LSs.

## Figures and Tables

**Figure 1 polymers-14-02834-f001:**
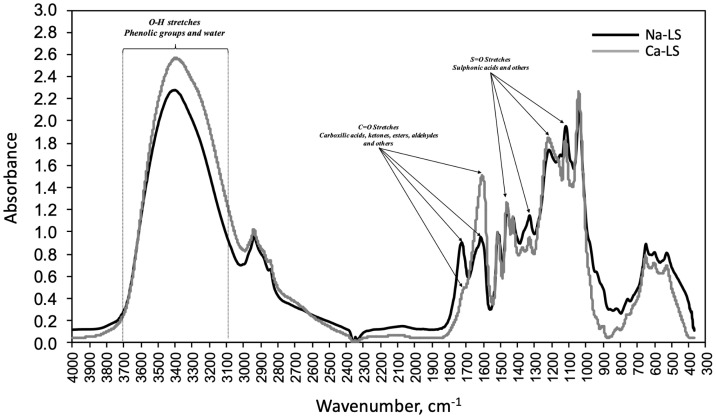
FTIR Spectra of samples Na-Ls and Ca-LS.

**Figure 2 polymers-14-02834-f002:**
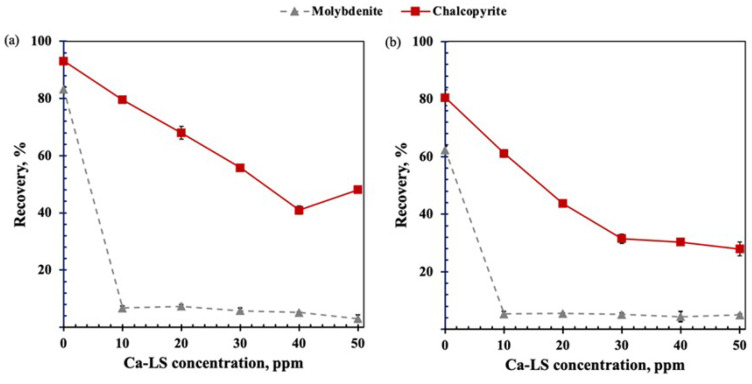
Depressant effect of Ca-LS on molybdenite and chalcopyrite in (**a**) 0.01M NaCl and (**b**) seawater, using 15 ppm MIBC and 25 ppm PAX as chalcopyrite collector at pH 8.

**Figure 3 polymers-14-02834-f003:**
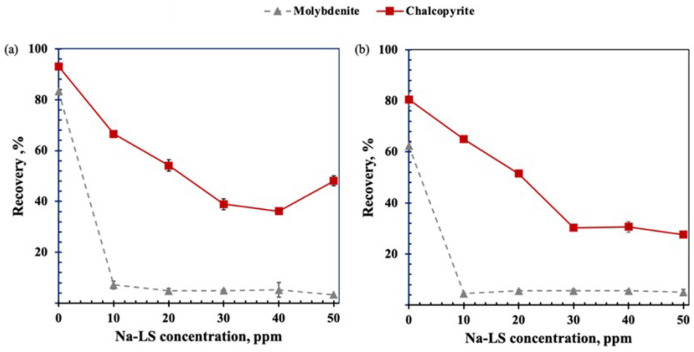
Depressant effect of Na-LS on molybdenite and chalcopyrite in (**a**) freshwater (0.01M NaCl) and (**b**) seawater, using 15 ppm MIBC and 25 ppm PAX as chalcopyrite collector at pH 8.

**Figure 4 polymers-14-02834-f004:**
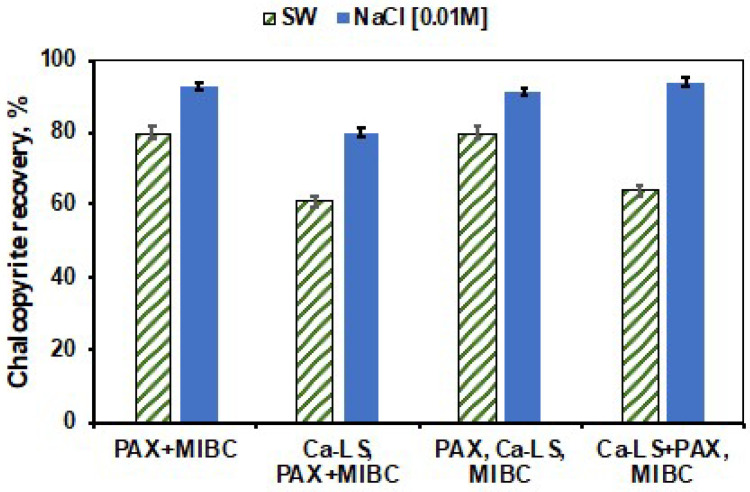
Effect of the order of addition of the reagents on the recovery of chalcopyrite at a concentration of 10 ppm Ca-LS, 25 ppm PAX and 15 ppm MIBC at pH 8 in seawater (SW) and 0.01 M NaCl.

**Figure 5 polymers-14-02834-f005:**
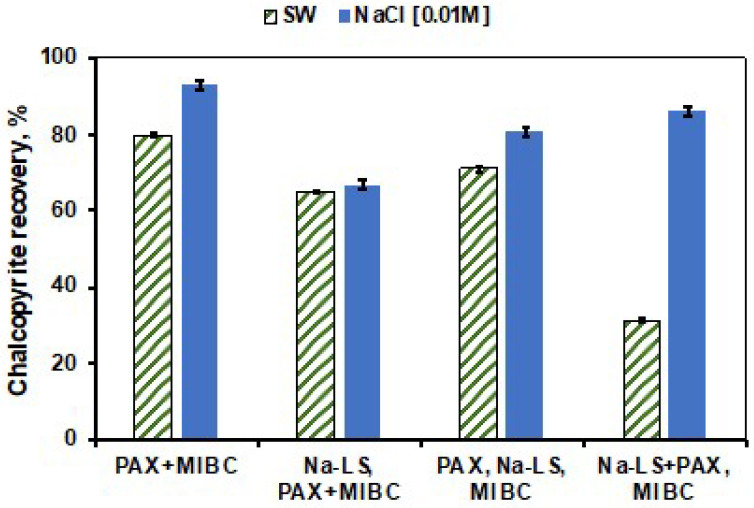
Effect of the order of addition of the reagents on the recovery of chalcopyrite at a concentration of 10 ppm Na-LS, 25 ppm PAX and 15 ppm MIBC at pH 8 in seawater (SW) and 0.01 M NaCl.

**Figure 6 polymers-14-02834-f006:**
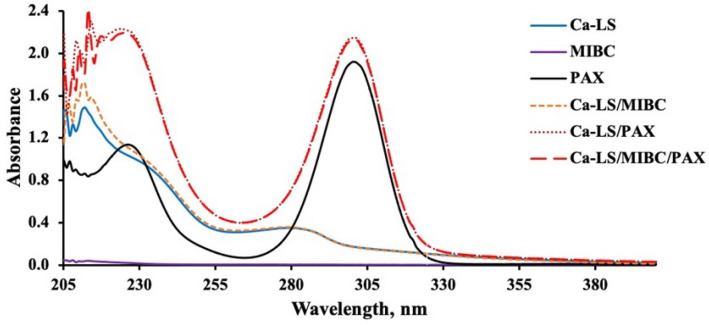
Absorbance spectra of Ca-LS (50 ppm), PAX (25 ppm) and MIBC (15 ppm), used in flotation without contacting the mineral in 0.01 M NaCl.

**Figure 7 polymers-14-02834-f007:**
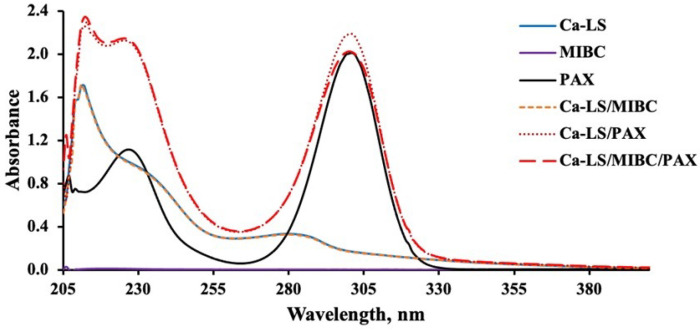
Absorbance spectra of Ca-LS (50 ppm), PAX (25 ppm) and MIBC (15 ppm), used in flotation without contacting the mineral in seawater.

**Figure 8 polymers-14-02834-f008:**
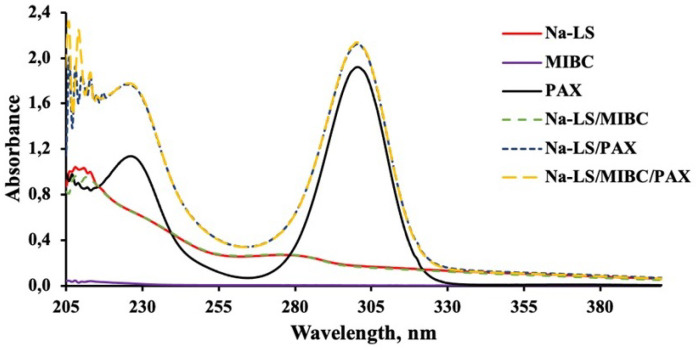
Absorbance spectra of Na-LS (50 ppm), PAX (25 ppm) and MIBC (15 ppm), used in flotation without contacting the mineral in (0.01 M NaCl).

**Figure 9 polymers-14-02834-f009:**
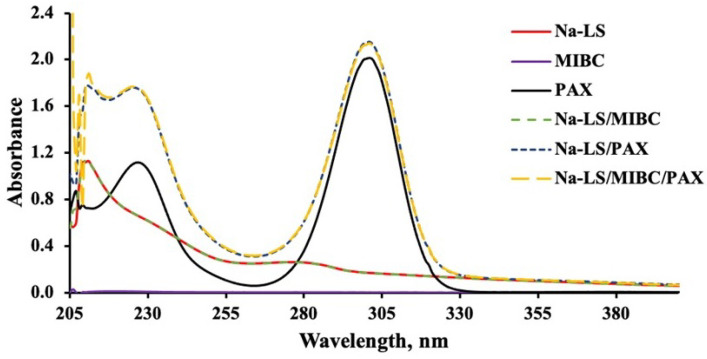
Absorbance spectra of Na-LS (50 ppm), PAX (25 ppm) and MIBC (15 ppm), used in flotation without contacting the mineral in seawater.

**Figure 10 polymers-14-02834-f010:**
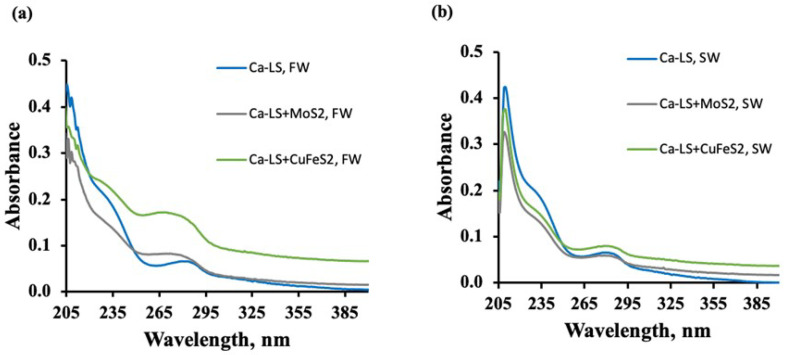
Absorbance spectra of Ca-LS (10 ppm) before and after contact with molybdenite and chalcopyrite in (**a**) freshwater (0.01 M NaCl) and (**b**) seawater.

**Figure 11 polymers-14-02834-f011:**
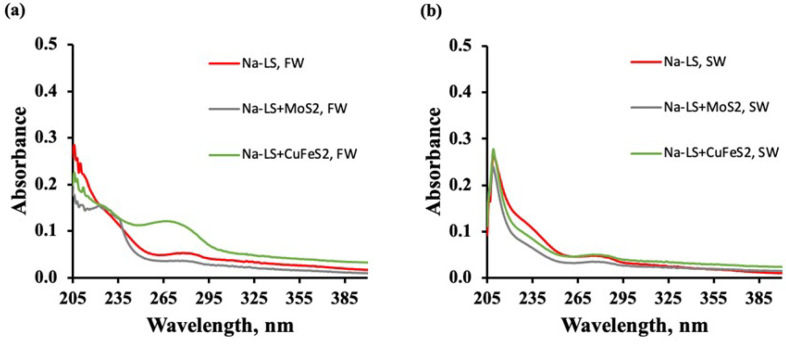
Absorbance spectra of Na-LS (10 ppm) before and after contact with molybdenite and chalcopyrite (**a**) freshwater (0.01M NaCl) (**a**,**b**) seawater.

**Table 1 polymers-14-02834-t001:** XRF analysis for chalcopyrite ore.

Chemical Element	(% w)	Chemical Element	(% w)
Cu	33.06	Na	0.43
Fe	29.33	Al	0.08
Si	0.64	K	0.02
S	29.20	As	0.02
Zn	0.09	Ti	0.01
Ca	0.14	Se	0.01
Mg	0.10	P	365 (ppm)
Ni	0.01	—	—

**Table 2 polymers-14-02834-t002:** XRF analysis for molybdenite concentrate.

Chemical Element	(% w)	Chemical Element	(% w)
Mo	45.76	K	0.08
S	39.87	Mg	0.05
Si	2.36	Cu	0.04
Al	0.28	Ti	0.01
Fe	0.20	Cl	0.07
Na	0.16	P	2788 ppm

**Table 3 polymers-14-02834-t003:** Order of addition of the reagents on the recovery of chalcopyrite using Ca-LS (10 ppm) and Na-LS (10 ppm), PAX (25 ppm), and MIBC (15 ppm).

Solvents	Order of Addition	Sequence ${}^{1}$	Cond. Time
Freshwater or seawater (pH 8)	Same time	Add Chalcopyrite	2 min
		Add lignosulfonate + PAX	10 min
		Add MIBC	5 min
	Separately	Add Chalcopyrite	2 min
		Add PAX	5 min
		Add lignosulfonate + MIBC	10 min

^1^ Concentrations of each reagent showed are the final concentrations in a total volume 110 mL.

**Table 4 polymers-14-02834-t004:** Equations of the calibration curves to measure the lignosulphonate concentration in seawater and freshwater.

Freshwater	Seawater
(2) $$\begin{array}{c} \begin{array}{c} Abs=0.0067[LS-Ca]+0.0068\\ R^{2}=0.9955 \end{array} \end{array}$$	(3) $$\begin{array}{c} \begin{array}{c} Abs=0.0065[LS-Ca]+0.0174\\ R^{2}=0.9992 \end{array} \end{array}$$
(4) $$\begin{array}{c} Abs=0.0050[LS-Na]+0.0053\\ R^{2}=0.9986 \end{array}$$	(5) $$\begin{array}{c} Abs=0.0049[LS-Na]+0.00283\\ R^{2}=0.9975 \end{array}$$

**Table 5 polymers-14-02834-t005:** Elemental analysis, molecular mass of lignosulfonates samples used in the study.

Analysis	Moisture%	C%	H%	N%	S%	Ca%	Na%	Mw Da	Mn Da	Mw/Mn
Na-LS	9.43	42.17	4.8	*˂*2.2	3.6	1.44	20.58	54,000	7268	7.43
Ca-LS	10.11	41.98	5.2	*˂*2.2	5.56	16.53	5.16	18,000	2500	7.20

**Table 6 polymers-14-02834-t006:** Zeta potential of Molybdenite and LSs adsorption on molybdenite in NaCl 0.01M and seawater. Using 10 ppm of LS.

	NaCl 0.01 M		Seawater	
LSs	PZ, mV	Absorbed LS, ppm	PZ, mV	Absorbed LS, ppm
None-LS	−46.60 ± 1.33	—–	−8.05 ± 0.54	——
Ca-LS	−30.92 ± 1.07	0.06	−11.08 ± 0.62	1.98
Na-LS	−36.55 ± 1.12	6.14	−10.07 ± 0.71	10.28

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
