# Peer review of "Understanding the Interaction of Lignosulfonates for the Separation of Molybdenite and Chalcopyrite in Seawater Flotation Processes"

_polymers, 2022, doi:10.3390/polym14142834_

Round 1
Reviewer 1 Report
Overall the paper is quite long and could be trimmed. The experimental section is roughly five pages long, and has a lot of detail . . . potentially too much detail and can be written more concisely.
In terms of the other comments:
Most Chilean copper mines tend to have a significant proportion (around 40 percent of the copper) as secondary copper sulphides (e.g. bornite, chalcocite) and are not predominantly chalcopyrite only.
In the third paragraph of the introduction on line 41 you use the work "impulsed". I think gave the motivation for this study is more appropriate.
In Table 2 the Fe assay is given twice.
The paragraph above Table 4 and Table 4 are a bit confusing. This needs to be simplified to say that the lignosulphonate was added before and after the Pax addition, and that the conditioning time for the lignosulphonate addition was 10 minutes in all instances.
In Section 2.6, Line 182 "by" should be replaced by "in".
The sentence starting on Line 215 ("The behaviour .. . . ") needs to be improved as it does not read well.
In Figure 4 you show how the order of addition affected copper recovery . . . what happened to the Molybdenum?
In line 296 you use "medias", medium is singular and media is considered to be the plural.
In line 315, I think the sentence should read: "In contrast, with Na-LS where the differences . . . "
The next two paragraphs are very hard to read as the English expression is very poor. Simplify.
While these experiments are interesting the proof would be in completing these tests in a mixed mineral system, then applying it to a real ore. Quite often when minerals are mixed together the resulting behaviour is not the same as the addition of single mineral results.
Author Response
Attached you will found the updated version of the manuscript according to the reviewer comments:
- Overall the paper is quite long and could be trimmed. The experimental section is roughly five pages long, and has a lot of detail. potentially too much detail and can be written more concisely.
R:/ You are correct, there was a mistake. According to the reference molybdenite is often associated with copper sulfides such as chalcopyrite (CuFeS2) and chalcocite (Cu2S). It was corrected. Please refer to the updated version of the manuscript.
- Most Chilean copper mines tend to have a significant proportion (around 40 percent of the copper) as secondary copper sulphides (e.g. bornite, chalcocite) and are not predominantly chalcopyrite only.
R:/ It was corrected. Please refer to the updated version of the manuscript.
- In the third paragraph of the introduction on line 41 you use the work "impulsed". I think gave themotivation for this study is more appropriate.
R/: It was corrected. Please refer to the updated version of the manuscript.
- In Table 2 the Fe assay is given twice.
R/: It was corrected. Please refer to the updated version of the manuscript.
- The paragraph above Table 4 and Table 4 are a bit confusing. This needs to be simplified to say that the lignosulphonate was added before and after the Pax addition, and that the conditioning time for the lignosulphonate addition was 10 minutes in all instances.
R/: It was corrected. Please refer to the updated version of the manuscript.
- In Section 2.6, Line 182 "by" should be replaced by "in".
R/: It was corrected. Please refer to the updated version of the manuscript.
- The sentence starting on Line 215 ("The behaviour .. . . ") needs to be improved as it does not read well.
R/: the sentence was eliminated. Please refer to the updated version of the manuscript
- In Figure 4 you show how the order of addition affected copper recovery . What happened to the Molybdenum?
R/: The order of addition did not affect the molybdenite recovery because the high depression caused by LSs and the collector is principally used to generated the hydrophobicity of chalcopyrite and to produce the flotation of this mineral.
- In line 296 you use "medias", medium is singular and media is considered to be the plural.
R/: It was corrected. Please refer to the updated version of the manuscript.
- In line 315, I think the sentence should read: "In contrast, with Na-LS where the differences . . . "
R/: It was corrected. Please refer to the updated version of the manuscript
- The next two paragraphs are very hard to read as the English expression is very poor. Simplify.
R/: The section of discussion was improved. Please refer to the updated version of the manuscript
- While these experiments are interesting the proof would be in completing these tests in a mixed mineral system, then applying it to a real ore. Quite often when minerals are mixed together the resulting behaviour is not the same as the addition of single mineral results.
R/: You are correct, this study is about the feasibility of use the commercial LSs evaluation to depress molybdenite in seawater to propose and alternative reagent to NaSH. After that, new research should be made to validate and propose the flotation process to separate chalcopyrite and molydenite in presence of this reagent. This comment was considered in the conclusion. Please refer to the updated version of the manuscript

Reviewer 2 Report
Comments from Reviewer
Title: Understanding the interaction of lignosulfonates for the separation of molybdenite and chalcopyrite in seawater flotation processes
The current form's presentation of methods and scientific results is satisfactory for publication in the Polymers journal. The minor and significant drawbacks to be addressed can be specified as follows:
1. Line 32. Lignosulfonate ----> Lignosulfonate(LS). See line 81. The abbreviation should be introduced earlier.
2. Line 42. Seawater ---> Seawater (SW). See Fig. 4 and figure captions. The abbreviation should be introduced earlier.
3. Line 84. 18,000? 2,500? In my opinion it is better to write “54000” and “7268”.
4. Fig. 1. (i) Tittle of y-axis? Units? (ii) Please use the colours for the lines.
5. Lines 99 and 100. The authors provided the full name of XRF. Now, consistently introduce XRD, FTIR, etc.
6. Line 150. UV-visible ---> UV-Visible.
7. Tab. 5. 0.005 ---> 0.0050.
8. I miss a table showing the ability of the tested materials in relation to other types of materials (better or worse).
Sincerely,
The reviewer.
Author Response
Attached you will found the updated version of the manuscript according to the reviewer comments:
- Line 32. Lignosulfonate ----> Lignosulfonate(LS). See line 81. The abbreviation should be introduced earlier.
R/: It was corrected. Please refer to the updated version of the manuscript.
- Line 42. Seawater ---> Seawater (SW). See Fig. 4 and figure captions. The abbreviation should be introduced earlier.
R/: It was corrected. Please refer to the updated version of the manuscript.
3.. Line 84. 18,000? 2,500? In my opinion it is better to write “54000” and “7268”.
R/: It was corrected. Please refer to the updated version of the manuscript.
- 1. (i) Tittle of y-axis? Units? (ii) Please use the colours for the lines.
R/: It was corrected. Please refer to the updated version of the manuscript.
- Lines 99 and 100. The authors provided the full name of XRF. Now, consistently introduce XRD, FTIR, etc.
R/: It was corrected. Please refer to the updated version of the manuscript.
- Line 150. UV-visible ---> UV-Visible.
R/: It was corrected. Please refer to the updated version of the manuscript.
- Tab. 5. 0.005 ---> 0.0050.
R/: It was corrected. Please refer to the updated version of the manuscript.
- I miss a table showing the ability of the tested materials in relation to other types of materials (better or worse)
R/: you are right, there are alternative reagents to depress molybdenite in conventional water. However, these reagents have not been tested in seawater.
Best regards,

Reviewer 3 Report
Manuscript entitled “Understanding the interaction of lignosulfonates for the separation of molybdenite and chalcopyrite in seawater flotation processes” submitted by Consuelo Quiroz, Romina Murga, Juan David Giraldo, Leopoldo Gutiérrez and Lina Uribe, can be considered for publication in Polymers Journal, after a major revision.
Here is a list of my specific comments:
1. General comment: The novelty and practical applicability of this study must be clearly highlighted in the manuscript.
2. Page 1, Abstract: Include in this section the most important results to highlight the importance of this study.
3. Page 2, line 35: “Considering that problematic…”. This paragraph should be detailed. The most important issues related to this topic should be briefly presented here.
4. Page 2, line 49: “The great challenge in the direct…”. This paragraph should be clearly reworded.
5. Page 2, line 60: “The objective of this research…”. At the end of Introduction, the main objectives of this study should be clearly and detailed presented.
6. Page 2, 2. Materials and Methods: This section should be systematized. In each subsection pay attention on the technical details and delete general observations/comments.
7. Page 6, 3. Results: In my opinion, this title should be replaced by “Results and discussion”. In addition, provide a clear and detailed discussion of the results immediately after their presentation. If no, all comments should be moved into next section.
8. Page 12, 4. Discussion: This section is too brief and should be detailed. All results presented in the previous section should be clearly discussed here in accordance with the main objectives of this study.
9. Page 14, 5. Conclusion: Include in this section the most important experimental results and findings to highlight the importance of this study.
10. Page 14, References: The number of references is too low and must be increased.
Author Response
Attached you will found the updated version of the manuscript according to the reviewer comments:
- General comment: The novelty and practical applicability of this study must be clearly highlighted in the manuscript.
R/: It was corrected. Please refer to the updated version of the manuscript.
- Page 1, Abstract: Include in this section the most important results to highlight the importance of this study.
R/: It was corrected. Please refer to the updated version of the manuscript.
- Page 2, line 35: “Considering that problematic…”. This paragraph should be detailed. The most important issues related to this topic should be briefly presented here.
R/: The problematic was detailed. Please refer to the updated version of the manuscript.
- Page 2, line 49: “The great challenge in the direct…”. This paragraph should be clearly reworded.
R/: This paragraph was corrected according to your comments. Please refer to the updated version of the manuscript
- Page 2, line 60: “The objective of this research…”. At the end of Introduction, the main objectives of this study should be clearly and detailed presented.
R/: It was corrected. Please refer to the updated version of the manuscript.
- Page 2, 2. Materials and Methods: This section should be systematized. In each subsection pay attention on the technical details and delete general observations/comments.
R/: It was corrected. Please refer to the updated version of the manuscript.
- Page 6, 3. Results: In my opinion, this title should be replaced by “Results and discussion”. In addition, provide a clear and detailed discussion of the results immediately after their presentation. If no, all comments should be moved into next section.
R/: It was corrected. Please refer to the updated version of the manuscript.
- Page 12, 4. Discussion: This section is too brief and should be detailed. All results presented in the previous section should be clearly discussed here in accordance with the main objectives of this study.
R/: It was corrected. Please refer to the updated version of the manuscript.
- Page 14, 5. Conclusion: Include in this section the most important experimental results and findings to highlight the importance of this study.
R/: This section was corrected according to your comments. Please refer to the updated version of the manuscript.
- Page 14, References: The number of references is too low and must be increased.
R/ Some additional references had been added. Please refer to the updated version of the manuscript.
Best regards,

Round 2
Reviewer 1 Report
Much better . . . the paper is still a little verbose, but the changes made are acceptable.
Writing clearly and concisely in English can be challenging . . . even for English speakers!
I think the paper is ready to go.
Author Response
Thank you for your comments.
Best regards,
Reviewer 3 Report
Manuscript entitled “Understanding the interaction of lignosulfonates for the separation of molybdenite and chalcopyrite in seawater flotation processes” submitted by Consuelo Quiroz, Romina Murga, Juan David Giraldo, Leopoldo Gutiérrez and Lina Uribe, can be considered for publication in Polymers Journal, after a major revision.
Here is a list of my specific comments:
- Page 2, line 38: “Previous work showed that commercial…”. Thos observation should be detailed.
- Page 2, 2.1. Reagents: This section should be reorganized. In this section, all the reagents used for these experiments should be presented. Move Figure 1 in Results section.
- Page 8, line 242: “In the Figure 4 is possible to observe…”. This paragraph should be reworded, and the expression “is possible to observe” should be replaced.
- Page 12, 4. Discussion: This section is too brief and should be detailed. All results presented in the previous section should be clearly discussed here in accordance with the main objectives of this study.
- Page 13, 5. Conclusion: This section should be reorganized. Delete the bullets and provide a clear presentation of the most important experimental results and findings included in this study.
- Page 14, References: The number of references is too low and must be increased.
Author Response
Dear reviewer, thank you for your comments. Attached you will find the updated version of the manuscript according to the comments:
- Page 2, line 38: “Previous work showed that commercial…”. Those observation should be detailed.
R:/ This idea was completed. Please refer to the updated version of the manuscript.
- Page 2, 2.1. Reagents: This section should be reorganized. In this section, all the reagents used for these experiments should be presented. Move Figure 1 in Results section.
R:/ This section was reorganized. Please refer to the updated version of the manuscript.
- Page 8, line 242: “In the Figure 4 is possible to observe…”. This paragraph should be reworded, and the expression “is possible to observe” should be replaced.
R:/ It was corrected. Please refer to the updated version of the manuscript.
- Page 12, 4. Discussion: This section is too brief and should be detailed. All results presented in the previous section should be clearly discussed here in accordance with the main objectives of this study.
R:/ This section was improved according to your comments. Please refer to the updated version of the manuscript.
- Page 13, 5. Conclusion: This section should be reorganized. Delete the bullets and provide a clear presentation of the most important experimental results and findings included in this study.
R:/ This section was reorganized. Please refer to the updated version of the manuscript.
- Page 14, References: The number of references is too low and must be increased.
R:/ New references were added. Please refer to the updated version of the manuscript.
Best regards,
